# RSOnet: An Image-Processing Framework for a Dual-Purpose Star Tracker as an Opportunistic Space Surveillance Sensor

**DOI:** 10.3390/s22155688

**Published:** 2022-07-29

**Authors:** Siddharth Dave, Ryan Clark, Regina S. K. Lee

**Affiliations:** Department of Earth and Space Science, York University, Toronto, ON M3J 1P3, Canada; rwclark@my.yorku.ca (R.C.); reginal@yorku.ca (R.S.K.L.)

**Keywords:** space domain awareness, resident space object, convolutional neural networks, multi-object tracking

## Abstract

A catalogue of over 22,000 objects in Earth’s orbit is currently maintained, and that number is expected to double within the next decade. Novel data collection regimes are needed to scale our ability to detect, track, classify and characterize resident space objects in a crowded low Earth orbit. This research presents RSOnet, an image-processing framework for space domain awareness using star trackers. Star trackers are cost-effective, flight proven, and require basic image processing to be used as an attitude-determination sensor. RSOnet is designed to augment the capabilities of a star tracker by becoming an opportunistic space-surveillance sensor. Our research demonstrates that star trackers are a feasible source for RSO detections in LEO by demonstrating the performance of RSOnet on real detections from a star-tracker-like imager in space. RSOnet convolutional-neural-network model architecture, graph-based multi-object classifier and characterization results are described in this paper.

## 1. Introduction

The number of resident space objects (RSO) in low Earth orbit (LEO) are expected to increase in the coming decade. There have already been several satellite–satellite collision(s), anti-satellite missile tests and satellite–debris impact events in the past decade. This places special emphasis on our ability to monitor, predict and mitigate the risks to space assets and human lives in Earth orbit. Space domain awareness (SDA) is the umbrella term for assessing the space environment, monitoring space assets, performing conjunction analysis, managing communication and navigation and custody operations [1]. The adequacy of our awareness is limited by our ability to collect qualitative and quantitative data of events in orbit. Given the wide scope of SDA, no single data-collection regime is effective at providing a comprehensive coverage of the space domain [2]. There have been numerous efforts around the world to enhance the current SDA capabilities. For example, the space surveillance network (SSN) is a U.S.-led group of instruments distributed worldwide with the objective to maintain a catalog of RSO population around Earth orbit. The SSN currently tracks, identifies and updates the trajectories of over 22,000 objects [3]. The SSN instruments vary by method, mode and platform in how they contribute to SDA [1]. J., April [4] summarized four key limitations with the current state of SSN: sensor count, geographical distribution, capability and availability. Most SSN instruments are expensive to build, maintain and replace, naturally reducing the numbers available. Many of these instruments have multiple objectives, and it is not uncommon for them to switch operation modes to meet non-SDA objectives. With increasing LEO activity, novel data-collection and processing techniques are needed to scale SDA operations.

In light of this, several feasibility studies evaluate data-collection regimes for SDA using instruments dedicated to non-SDA mission objectives. This paper focuses on an existing low-cost optical instrument known as star tracker. A star tracker is a commonly used imaging instrument on satellites for attitude determination. A star tracker takes periodic images known as star-field images. Using a catalogue of stars stored in memory, an automated algorithm estimates the orientation of the satellite based on the position and orientation of the matching stars in the image sequence [5]. The star tracker’s imaging priority is to observe stars, and any other objects (Earth limb, moon, etc.) captured by the imager is considered noise. Non-star objects detected by the star tracker, which are otherwise filtered, present an opportunity to collect novel data valuable for SDA, as a dual-purpose star tracker. Compared to the 15 cm aperture telescope such as on the Canadian SDA satellite NEOSSat [6], a star tracker would be a less qualitative SDA sensor. However, given the number and geographical spread of potential dual-purpose star trackers, and the likelihood of frequent and repeated overlapping (multi-site) observations, a virtual constellation of star trackers would be a more quantitative SDA sensor [7]. It is difficult to survey an exact count, but it can be assumed that there are several thousand star trackers already in LEO. For example, it is known that each of the over 2000 SpaceX Starlink satellites currently in orbit carries at least two star trackers [8]. The orbital dynamics of such a system will be discussed in future works, whereas this paper presents an image processor, RSOnet, as a framework for processing star-field images.

The objective of this research is to build on previous feasibility efforts by demonstrating an example star-field image processor, referred to as RSOnet, for a dual-purpose star-tracker concept. Our methodology focuses on integrating space surveillance requirements into the existing star-field image-processing framework. Requirements such as detection, tracking, classification, characterization and data compression ratio are selected as evaluation metrics. This paper describes the design, performance and applications of RSOnet. This paper also describes the benefits of using RSOnet as a tool to amplify quantitative decentralized data collection to enable further research.

## 2. Relevant Works

### 2.1. Dual-Purpose Star Tracker

Clemens performed a study to determine the feasibility of RSO detections from commercial off-the-shelf (COTS) star-tracker cameras. Clemens’ preliminary results, using an optical image simulator, found that a baseline of 1 to 10 RSO detections per day is achievable [9]. Clemens’ research was verified and validated using the Fast Auroral Imager (FAI) on the Canadian satellite CASSIOPE. Compared to the 0.8${}^{\circ }$ field of view (FOV) of NEOSSat, the FAI has a 26${}^{\circ }$ FOV, whereas COTS star trackers range between 8${}^{\circ }$ and 45${}^{\circ }$ depending on the manufacturer and application. Star trackers generally have a wide FOV, small aperture and capture images with a short exposure time to capture light from many bright stars and analyze their motion on the image plane. The FAI [10] is a similarly tasked instrument, with a comparable aperture of 17 mm, and is also considered a primary data source for RSOnet. In addition to the FAI, data for developing RSOnet was also sourced from the space-based optical image simulator (SBOIS) developed by Clark [11]. The validation data obtained from real FAI images are labeled manually and not used for training. Part of the labeling is performed using the Astrometry.net plate-solver [12]. Labeled training and validation data obtained from simulated SBOIS images, of the matching FAI scenario, are used both for training and validation.

### 2.2. Image Processing

Research by Denver [7] studied the sensitivity performance of a star tracker for debris detection. Fundamentally, detection of RSO using satellite-mounted camera improves based on two key parameters, aperture size and the object-detection sensitivity of the image-processing algorithm. For RSO detection, Denver estimates limiting visual magnitudes of 7–9 is possible with star trackers and Clemens estimates a limiting visual magnitude of 8.7. Building on this, RSOnet is designed as a framework to optimize for object detection, characterization and classification. An object in RSOnet includes physical and virtual objects in an image. Physical objects are celestial objects and Earth-orbiting satellites and virtual objects are hot pixels and shot noise. In the context of typical operation mode in a small aperture star tracker providing attitude updates at rates of up to 5 Hz, the objects detected can be classified spatio-temporally. The motion and brightness of each object over time is key to classifying the object class. Badura’s [13] implementation of CNN demonstrates how recent advances in artificial-intelligence techniques can be used for SDA applications, by classifying objects and their attitude status with light curves from images. It is also clear that RSO position and velocity can be estimated [7,14]. The accuracy of the RSO position and velocity estimation, which is likely unreliable using just one star tracker, can be improved using a network of star trackers and will be addressed in future works. Other object-detection techniques such as support vector machines (SVMs) are also popular in astronomical image processing, for both classification and regression applications [15]. Before artificial-intelligence and machine-learning techniques, star detection and centroiding was commonly performed using Gaussian fitting techniques [16]. In this method, manually designed variations in Gaussian distribution functions were used as filters, and the filter activations were used by a generalized model function to predict a detection probability and perform regression-like analysis [17]. In this research, a blend of Gaussian distribution functions are used to generate training data, a CNN backbone is trained and validated on a mixture of real and simulated images. A similar effort is described in previous works, where a CNN is used to model point spread function for astronomical applications [18].

A key step after processing detections is multi-object tracking (MOT). In star-field images, multiple point-source objects have varying angular velocities, which is a key feature for classification. The FAI images have a 0.1 s exposure, avoiding a scenario where a moving point-source object creates a streak. Therefore, sequenced over many image frames, an object can be tracked and differentiated using the MOT approach. Techniques such as image stacking to reduce noise and Gaussian parametrization to determine streak characteristics and object tracking have been studied in the past [19,20] and also utilized for faint satellites in geostationary Earth orbit [21]. These techniques are not suitable for wide star-field images, as active tracking, ranging and long exposure images are not in the scope of this research. Instead, tracking and classification techniques such as re-identification [22] and shortest path are considered [23].

### 2.3. Space Domain Awareness and Applications

RSOnet combines photometric and astrometric feature extraction of objects that are beneficial in estimating RSO attitude and spin rate [24], RSO identification [25], host-satellite attitude determination and potentially more applications. All such applications require a large labeled and continually updated training data set of star-field images, something not currently available for public researchers. RSOnet provides a framework for a scalable and portable tool for star-field image processing and data collection. On-going efforts at York University as a part of the Canadian-Space-Agency-funded FAST grant award to launch a star-tracker design on a stratospheric balloon mission aims to demonstrate the portability of RSOnet. This project aims to collect and process thousands of images from a high-altitude payload to detect RSO for SDA applications. During the suborbital mission, RSOnet will also be used as a tool for image data compression, demonstrating the tool’s advantages in the context of data collection, storage and downlink.

## 3. RSOnet: Star-Field Image Processing

This section describes the approach adopted for processing image sequences. Firstly, a CNN architecture is described along with the training process. Secondly, the processed frames are used to generate a network graph for tracking. Finally, the object tracks are used for classification and optical characterization.

### 3.1. RSOnet: CNN Architecture and Training

In modern space applications, optical sensors based on CMOS or CCD technology are widely popular. Both work on the principle of photon accumulation in individual pixel bins read out by a series of electronic circuits. This produces a digital image such as a star-field image. When a satellite-mounted imager is exposed to the dark sky through a telescopic lens, several sources of light may be present. Filtering for the point spread function (PSF-shaped) objects leaves celestial bodies, RSO, and noise. Given the wide field of view characteristic of star-field images, each PSF-shaped object fits in a bounding box square with a side of 7 pixels. The 7 pixel specification is applicable to the FAI images in particular; however, image binning and pooling layers provide the needed flexibility for higher resolution images as well. The uncertainties of PSF modeling can impact the characterization of objects that are detected in the image. A source of error during the imaging process is the PSF of the camera.

The object size constraint provides the upper limit for the filter size, and defines the macro image or macro. The size constraint also presents a novel training opportunity for the CNN which removes the dependence on a real labeled data set by using the Monte Carlo method. In this case, a random PSF generator is paired with an optical instrument simulator for creating training images. A synthetic macro is a 7-pixel square image containing a PSF-shaped object with predefined characteristics. Previous efforts have described a similar process [18]. Figure 1 shows examples of real, simulated and synthetic macros for a visual comparison.

The synthetic macro generator (SMG) uses the Monte Carlo method to iterate over predictable parameters such as position, brightness, angular size, angular velocity, and background noise. The SMG bins the data based on image resolution, pixel bit resolution, exposure time, quantum efficiency, field of view, macro size, read noise, and thermal noise, all of which are parameters of the optical sensor, in this case, the FAI. A corresponding CNN architecture was designed to accompany the SMG to create a tool with the ability to adjust training parameters shown in Figure 2.

The CNN for RSOnet consists of three convolutional layers followed by a fully connected layer. A convolutional layer is composed of convolution filters that, when applied to a training data set, assist in training the spatial dependencies of objects. A fully connected or dense layer is a feed-forward neural network that is fed the output of the convolutional layers to train for patterns in convolution filter activations for identifying objects in an image. The CNN architecture is shown in Figure 3. The FAI image size is the input of size 256×256×1. After the first convolution, batch normalization and pooling step, the dimensionality is 128×128×64, where 64 is the number of filters. By the third convolution step, enough contextual information is summarized by the 128 filters in each region. At this stage, each region is processed for detection using a fully connected layer with a classification and regression head for detection and localization.

The output of the neural network was split for classification, classification confidence and regression using softmax, sigmoid and rectified linear activation functions, respectively. The two classes are detection and non-detection. Regression involved prediction of PSF features provided to SMG for image creation. The key outputs from processed images are features of a digitized 2D Gaussian distribution such as mean, standard deviation and photon count, which are utilized by the SMG. The results of this training are shown and discussed in Section 4.

### 3.2. Tracking and Classification

Detected objects are then classified into stars, RSO or types of noise during the tracking process. In wide-field-of-view star-field images, there can exist hundreds of potential detections with varying angular motion. Groupings of angular motion separate stars from RSO. To perform multi-object tracking (MOT), detections from a processed image sequence are used to create a k-partite graph layout (directed acyclic graph), where k is the total number of images processed [26,27,28]. In this layout, each detection acts as a node and no edges connect detections from the same frame. There is a start node, end node and a detection node. The detection node is further split into nodes u and v. An example graph is illustrated in Figure 4. The graph is then simplified by using a distance limit of 20 pixels between two frames, as an upper bound on the angular velocity, to reduce required computation.

The objective function, *T*, is described below in Equation (Equation 1). There are four cost functions associated with the objective function to track the motion of an object detected in multiple consecutive frames of a sequence. The first function is the cost of a tracking edge, defined by $C_{t}$, which is based on polynomial curve fitting metrics, with a maximum polynomial order of 1. The cost of the tracking edge depends on the distance d traveled between the two nodes of the edge and the variation in the brightness b of the object. The second and third functions are the cost of starting a new object track and track termination, as defined by $C_{in}$ and $C_{out}$, respectively. Finally, $C_{det}$ defines the cost of detection, which is the negative cost associated with detection probability. $B_{i}$ refers to the probability that the detection is false. *f* defines the corresponding indicator function, which identifies which nodes are connected by taking on a binary value of 0 or 1. Starting and ending cost functions provide connectivity to all nodes of the graph to ensure tracks can be started and ended at any point during the sequence due to detection discontinuity. Both the start and end cost functions are designed to preserve track flow conservation and are a fixed value. Flow conservation ensures that the inflowing indicator function adds up to the out-flowing indicator function, which is between 0 and 1. This allows the graph solving algorithm to determine the lowest cost approach to start, traverse and end object tracks.
(1)$$\begin{array}{c} T=argmin\sum_{i}C_{in}\left(i\right)f_{in}\left(i\right)+\sum_{i,j}C_{t}\left(i,j\right)f_{t}\left(i,j\right)+\\ \sum_{i}C_{det}\left(i\right)f_{det}\left(i\right)+\sum_{i}C_{out}\left(i\right)f_{out}\left(i\right) \end{array}$$
(2)$$\begin{array}{c} C_{det}=log\frac{B_{i}}{1-B_{i}} \end{array}$$
(3)$$\begin{array}{c} C_{t}=\sum_{i,j}d\left(i,j\right)+b\left(i,j\right) \end{array}$$

The track of an object is classified into celestial objects, RSO and noise.

## 4. Evaluation Metrics and Analysis

Evaluation of the CNN and training process is demonstrated with precision and root mean squared error (RMSE) during the training process. A total of 50 epochs of training were performed to obtain high precision for the training data and a significant improvement against the validation data. Stopping training at 50 epochs avoided over-fitting to the synthetic data set, which, in turn, yielded better performance against simulated and real data sets. Figure 5a,b illustrates the training progression. RMSE values for regression achieved the desired PSF mean accuracy of 0.02 by epoch 30. Several combinations of the SMG, SBOIS and real FAI data set were attempted for training. The final training data were obtained from the SMG and SBOIS [11], at a shuffled split of 80–20. The overall training and validation data split was also at 80–20. Approximately 7 million object instances were used during training and validation with varying noise levels, object brightness, object PSF-shapes, and motion vectors.

The accuracy of the CNN is measured against synthetic, simulated and real images for comparison. Object detection accuracy is measured by looking at the ratio between the sum of true positives and true negatives to the sum of all four possible outcomes, only for objects with an SNR above 6. As expected, the detection accuracy of the network drops against simulated and real images. The accuracy of detection against the real data set can certainly be improved provided a real labeled data set is folded into the training process. Object characterization is demonstrated with mean squared error (MSE) and RMSE for mean, standard deviation and photon count against synthetic and simulated data. The regression accuracies of the PSF mean or object centroid, PSF scale or object angular size and photon count have significantly better performance with the synthetic data. This drop in accuracy is reflective of the apparent differences in the three data sets considered. The accuracy values for each data set are also limited by the labeling method. In the SMG training images, faint objects that might otherwise be undetectable generate a false flag, throwing off the accuracy. This is persistent even for images from SBOIS. Additionally, in the real FAI images, the RSO labels are highly inconsistent and may not include fainter RSOs that would be detectable.

The PSF modeling effort in [18] yielded a PSF mean of $0.784\pm 1.561\times 10^{-3}$ and $0.931\pm 1.987\times 10^{-3}$ for the x and y axis, respectively. RSOnet achieves a marginally better accuracy; however, the results are not directly comparable. The training-data parameters, their range and the data set used for training and validation are not identical. However, a similar training process, using SMG, generated nearly equivalent results. Furthermore, the PSF scale, or full-width half max (FWHM), for [18], is marginally better than RSOnet at $0.2655\pm 11.081\times 10^{-3}$. The likely factor is that RSOs considered during the training-data generation process have an angular velocity component, which directly affects the network’s ability to resolve the true angular size.

When compared to the PSF-fitting method, the CNN designed and trained in this study resolved images 44% faster. The PSF-fitting method achieved a centroiding accuracy of 0.17 on real images, which is marginally better than RSOnet. However, the biggest advantage of RSOnet is the SNR cutoff, which was defined at 6 dB compared to 18 dB for PSF fitting for this data set. PSF fitting has demonstrated 4 dB SNR on another astronomy data set [29]; however, RSOnet was trained using a custom data set generated by the SMG specifically built for the FAI sensor. The combined SMG and RSOnet training process ties hardware specifications to algorithm performance in wide-FOV camera sensors, and also makes it a transferable and modular framework. Figure 6, below, shows the filter responses for objects detected on a set of real FAI images, one with a lens flare. Figure 6, below, and Table 1, above, demonstrate the resiliency of the RSOnet CNN to detect faint objects and handle noise real images due to synthetic data training.

For evaluating object class identification, the accuracy of each class is shown in Table 2 from the graph-based tracking results. RSOnet’s graph-based MOT implementation shows promising classification results and can certainly be improved with real data. Previous efforts use a stacking or streak characterization approach for classification problems. The closest comparison is made to [30], which studies GEO object tracks over long exposure images.

Compared to [30], RSOnet-derived MOT implementation achieves a lower classification accuracy and precision, but a comparable recall value. In the compared literature, GEO objects traverse the imager’s field of view at a slower speed than for RSOnet’s FAI images. Additionally, the algorithm has to differentiate between stars, which are small streaks, and GEO objects, which are long streaks. In the case of FAI and RSOnet, the algorithm has to first generate the best tracks to append together into a comparable tracklet. The track proposal is based on shortest distance; however, that is not always accurate given eclipse scenarios. Despite differences in image type, data set and algorithm, the performance of RSOnet demonstrates classification for a multitude of objects from low-exposure, high-frame-rate image sequences. Further improvements based on more accurate labels can improve RSOnet’s classification accuracy.

It is far easier to track and classify stars than RSO because of their coordinated motion and non-fluctuating visual magnitude over the course of an image sequence. RSO, on the other hand, tend to cross paths with other RSO, stars and noise, making it harder to classify RSO accurately. In terms of classifying noise sources such as hot pixels, shot noise, Earth’s limb and the moon, the vaguer definition of noise for the CNN makes for a more difficult classifier. Hot pixels are the easiest amongst the noise to classify because the same pixel is being activated continuously throughout an image sequence. In future work, attempts should be made to tie real verified detections using an orbital propagator. In this method, the RSO catalog information is used to pair a detection to an orbital track to verify if the RSO sequence detected by the imager is a true positive [31,32].

## 5. Discussion

The PSF mean accuracy of RSOnet is outlined in pixels and not acrseconds, to provide the adaptability to various star-tracker-like imagers. For the 256 × 256 pixels of an FAI image, an accuracy of 0.2 pixels represents 73 arcseconds. In contrast, a similar telescope with a higher resolution imager, such as a wide FOV camera, at 1024 × 1024 pixels could represent an accuracy of up to 18 arcseconds. Similarly, the detection sensitivity of RSOnet on FAI image sequences is 9. A magnitude of 9 is not enough to detect objects in geostationary orbit, but is sufficient for objects less than 2000 km away in LEO. As predicted, these are qualitatively a worse measurement than the NEOSSat imager with 2.3 arcseconds accuracy. This can largely be attributed to the smaller aperture size and worse image resolution on the FAI. Star trackers have a narrower FOV and better image resolution comparable to FAI. Additionally, quantitative multi-site observations have the potential to reduce the overall accuracy of the RSO astrometry using joint probabilistic data association (JPDA) [1], which will be explored in future works.

### Simulated Orbit Determination Scenario

A simulation scenario was created for this study to assess how star-field images processed with RSOnet could then be used for orbit determination efforts. The scenario was created using SBOIS [11], which consisted of a ground and a space observer viewing the 3 RADARSAT constellation mission (RCM) satellites, as illustrated in Figure 7, below. The ground and space camera has specifications identical to that of the FAI, and the space-based observer adopts CASSIOPE’s orbit. The ground observer’s location is selected to be where it can observe RCM at the same time as the space observer. This defines a multi-site scenario with just one space observer, demonstrating potential for observation overlaps between individual star trackers and established ground observers. In Figure 7, there are six nodes that define the scenario: Earth *E*, ground observer *G*, space observer *S*, and RCM 1, 2 and 3 as R1, R2 and R3, respectively. The auxiliary data from SBOIS includes position vectors for each node at each image frame and follows the naming convention $P_{node}$. The unknowns are defined unit vector estimates from each observer as $p_{terminal\;nodes}$.

Figure 8, below, highlights the key parameters required from the space and ground observer to determine the unit vectors estimates for $p_{SR1}$ and $p_{GR1}$. Here, $x_{i}$, $y_{i}$ and $z_{i}$ represent the position of the host spacecraft $P_{S}$ at time instance *i*. The spacecraft attitude $Q_{S}$ is defined by the yaw, pitch and roll at each time instance. $E_{S}$ represents the time-independent mount offset to the boresight of the imager relative to the body frame of the spacecraft. $\delta_{SR1}$ represents the angular displacement from the boresight for R1, after accounting for the spacecraft roll. Finally, $\alpha_{SR1}$ defines the cross-track accuracy error of individual measurements.

From the body frame reference of the spacecraft, the attitude and imager-mount offset are added to produce the imager boresight unit vector, as shown in Equation (Equation 4).
(4)$$\begin{array}{c} p_{boresight}=p_{S}+R\left(E_{S}+Q_{S}\right) \end{array}$$
where the function R(ϕ) is the rotation applied to the unit vector referenced to the satellite body frame. At each time step i, the imager boresight unit vector is computed. Using the localization results from RSOnet, each RSO is recorded as an angular displacement vector $\delta_{SR1}$ and cross-track localization error $\alpha_{SR1}$ referenced to the boresight. Finally, at each time step, the origin of the unit vector $p_{SR1}$ is assigned as the spacecraft position vector $P_{S}$.

Two independent unit vectors and cross-track measurement errors are obtained from each observer. The closest point between the two unit vectors is used to determine the missing range component, which is used to convert the angular cross-track error from arcseconds into kilometers. The RCM scenario contained six independent image sequences, each consisting of 27–43 images set 1 s apart from both the ground and space observer. The averaged results for this scenario are summarized in Table 3 below. The cross-track error is presented in meters as the range values are consistent for the image sequences, approximately 1100 and 1700 km for ground and space observers, respectively. The average range error to target is computed based on the closest approach point between unit vectors $p_{SR1}$ and $p_{GR1}$. Finally, the average displacement is computed based on range estimates for each observer and the combined observation for all image sequences.

Each position estimate for the RCM satellite is verified against the simulator’s high-accuracy position. On average, each RCM position estimate falls within 2 km of the actual position. There are three key factors that affect the cross-track accuracy of an observation: observer position, attitude and cross-track localization accuracy from RSOnet. Each vary in their effect on the final position estimation depending on range to target and relative cross-track angular velocity. Factors other than the image processing can certainly affect the accuracies of the measurements conducted using the star-field images from a star tracker. Improving image resolution, high-accuracy host-spacecraft positioning and attitude determination, and increasing the number of observers improves position estimation accuracy.

## 6. Conclusions

In conclusion, this paper describes the design and verification of RSOnet as capable of using wide-FOV camera sensors in a dual mode of operation. RSOnet has demonstrated the ability to detect, track and classify RSO based on their angular motion and this data is currently being implemented for further research. RSOnet successfully performs image data compression by storing only the key bits of information. Depending on the size of the image and the number of objects detected, RSOnet compression ratios range from $2.4\times 10^{-4}$ to $3.1\times 10^{-2}$. With these compression ratios, the cost of data downlink is significantly reduced, making it simpler to create a data repository than before.

This paper also addresses how position estimates can be derived using multiple observers, and highlights the key factors that affect the positioning accuracy. Accumulated over several passes, position and velocity estimates of RSO can augment orbit determination efforts which are a key aspect of SDA. By activating a network of star trackers, big and small, the downlinked data can be centrally processed for heavy computational tasks such as orbit prediction, determination, conjunction analysis and calibration efforts.

A key benefit of RSO characterization on star trackers relate to astrometry and photometry applications. Many long-duration sequences of LEO RSO are possible when observing from space, something not possible from a fixed ground-based sensor. The long duration sequences benefit both astrometry and photometry in their ability to resolve ambiguity. From space, long duration photometry is easier due to the lack of atmospheric distortion. Photometry technique such as light curve analysis has the potential to estimate shape and spin rates of target RSO, and can also aid in satellite identification and remote assessment. Although most photometric analysis is performed on a higher class of telescopes compared to star trackers, a network of star trackers provides the unique opportunity of observing the same target RSO from various locations at once and/or consecutively. Accumulating and discovering such situational data has the opportunity to advance SDA research in the academic community. The next phase of RSOnet development leads to a hardware accelerated implementation on a technology demonstration mission.

## Figures and Tables

**Figure 1 sensors-22-05688-f001:**
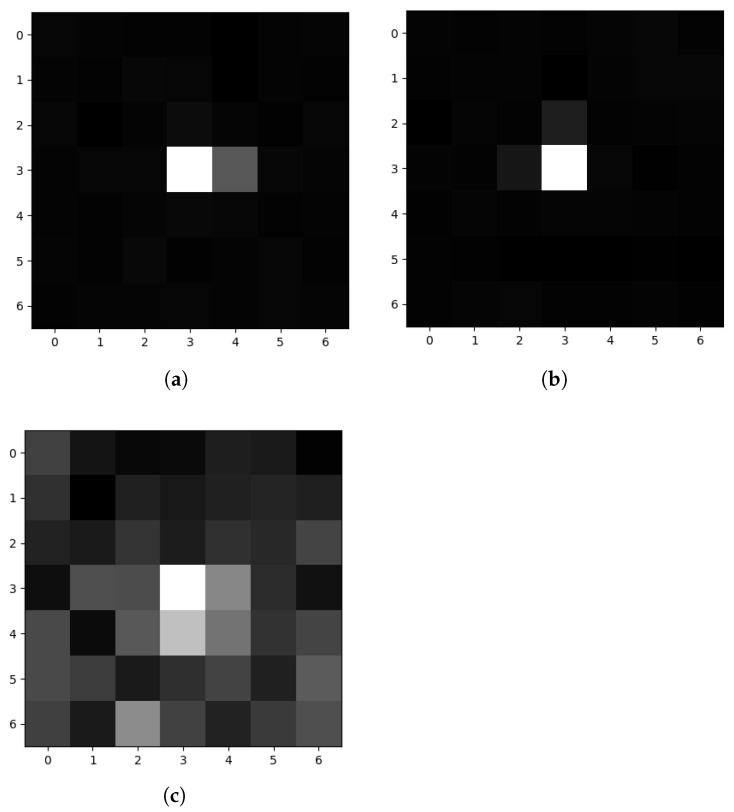
Examples of raw macro images. (**a**) Synthetic; (**b**) Simulated [24]; (**c**) Real [10].

**Figure 2 sensors-22-05688-f002:**
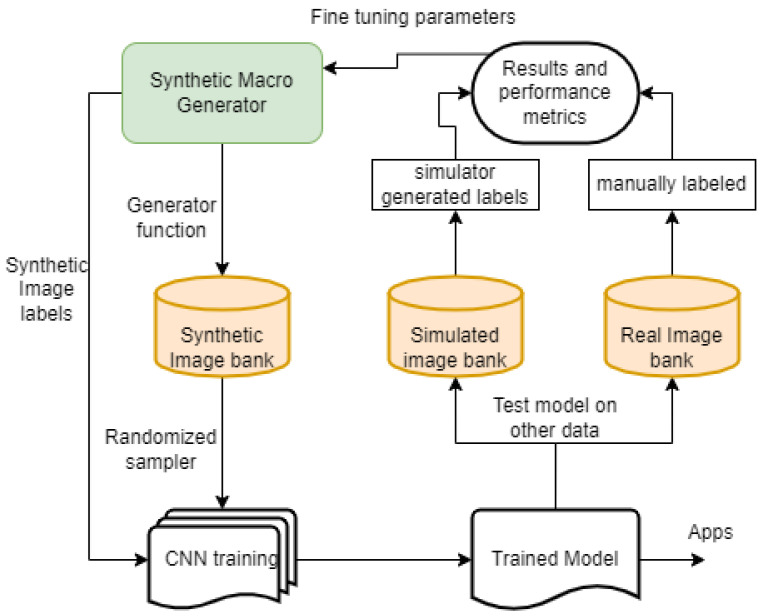
CNN training method.

**Figure 3 sensors-22-05688-f003:**
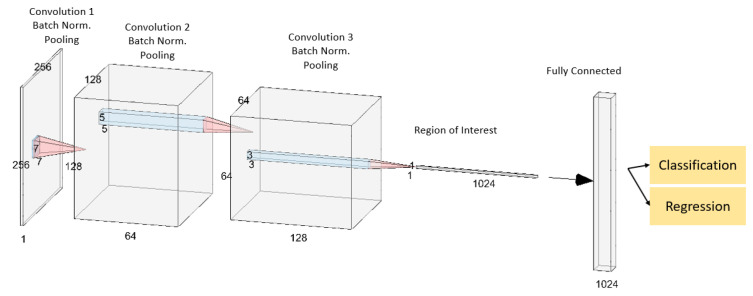
CNN architecture and its outputs.

**Figure 4 sensors-22-05688-f004:**
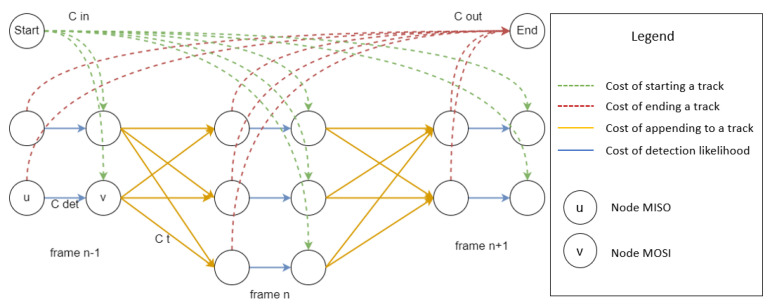
Multi-partite graph layout for multi-object tracking.

**Figure 5 sensors-22-05688-f005:**
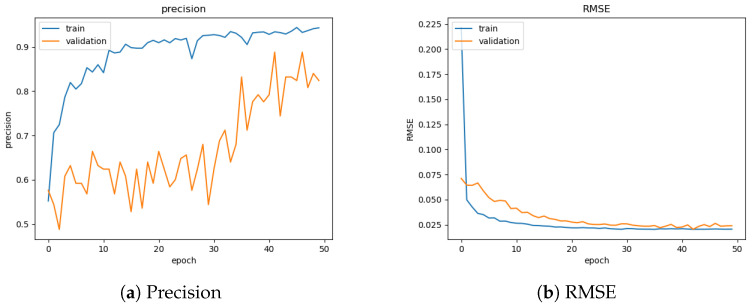
CNN training evaluation.

**Figure 6 sensors-22-05688-f006:**
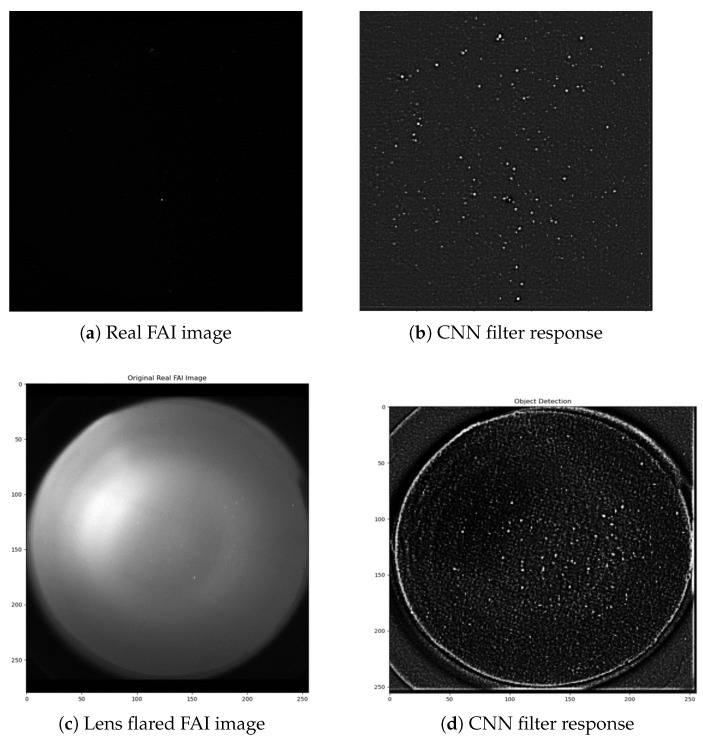
RSOnet on real FAI images.

**Figure 7 sensors-22-05688-f007:**
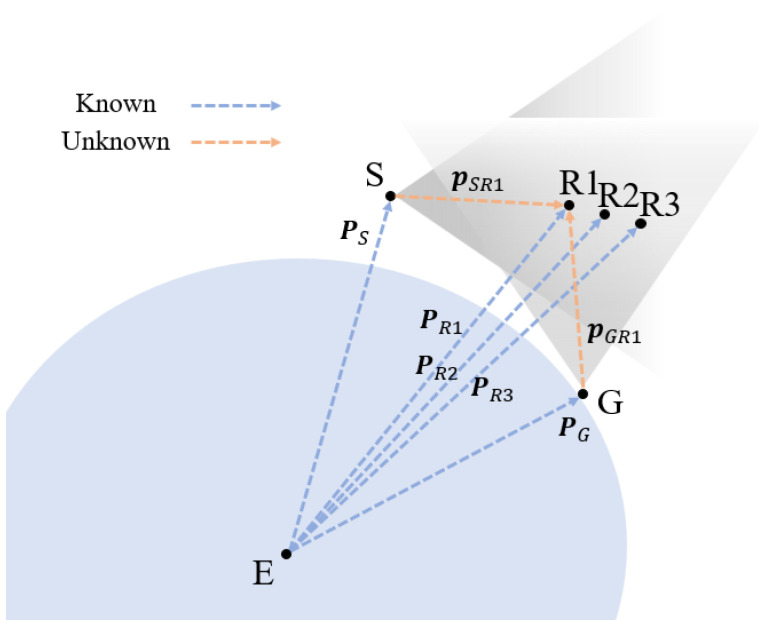
Illustration of simulated multi-site observation of RADARSAT constellation mission.

**Figure 8 sensors-22-05688-f008:**
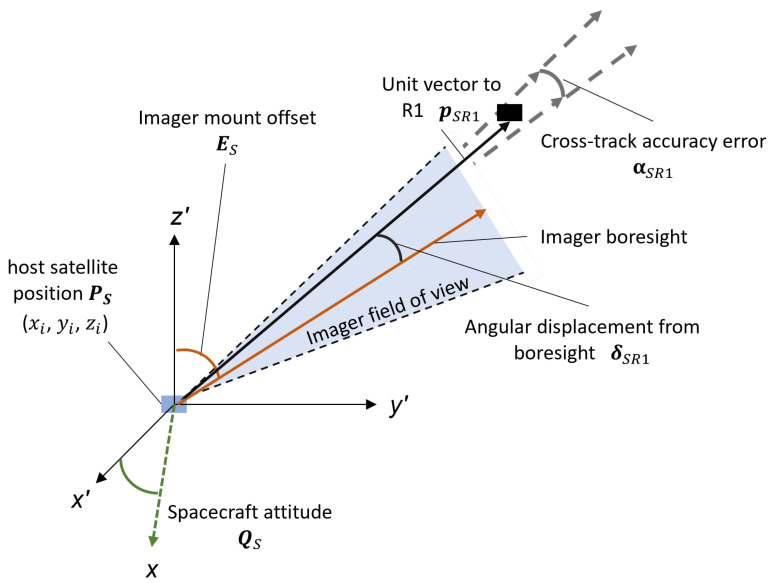
Observer’s reference frame and definition of key astrometric parameters.

**Table 1 sensors-22-05688-t001:** CNN accuracy results after 50 epochs against three data sets.

Function	Synthetic	Simulated	Real
Object detection	92%	83%	88%
PSF mean (MSE)	$5\times 10^{-4}$	$4\times 10^{-2}$	0.2
PSF scale (RMSE)	$4\times 10^{-2}$	0.11	0.15
Photon count (RMSE)	$7\times 10^{-2}$	0.1	0.1

**Table 2 sensors-22-05688-t002:** Tracking and classification performance metrics.

	Star	RSO	Noise
Accuracy	84.2%	68.1%	32.6%
Precision	81.1%	60.4%	25.8%
Recall	98.5%	95.6%	30.0%

**Table 3 sensors-22-05688-t003:** Error in RSO position estimates for RCM scenario.

	Cross-Track (m)	Range (m)	Displacement (m)
Spacecraft	955	N/A	2086
Ground	683	N/A	1751
Combined	N/A	1215	1512

## Data Availability

The data presented in this study are available on request from the corresponding author. The data is not publicly available due to continuing research and containing sensitive data.

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
