# Peer review of "RSOnet: An Image-Processing Framework for a Dual-Purpose Star Tracker as an Opportunistic Space Surveillance Sensor"

_sensors, 2022, doi:10.3390/s22155688_

Round 1

Reviewer 1 Report

The implied assumption that LEO objects can have their orbit ephemeris determined by star tracker sensors is not correct. Precise ephemeris can only be done by active interrogation.

If the authors change their methodology as written in the reviewer's report, a rewritten manuscript would be considered.

Author Response

This paper has the wrong methodology. It is not possible to derive a satellite’s state vectors (6 degrees of freedom: 3 position and three velocity) at an instant of time using star tracker sensors (azimuth and elevation provide insufficient accuracy). Precise ephemeris (the only type that has value) determination can only be done with active interrogation. Discussion of debris fields is the wrong physics.

The research conducted in this paper does not argue that state vectors or orbit determination is possible with an image at one instance of time. Instead the paper attempts to describe how a star field image processing algorithm can detect stars and RSO in the image, track and classify them for future applications. Part of these future applications are enabled by this algorithm to collect and compress data in orbit from thousands of star trackers to form a more real-time holistic situational awareness of the objects in Earth orbit.

What is the correct physics using star trackers in an expanded role? Self-defense against parasitic satellites (para-sats)
is the right physics. Geosynchronous satellites are particularly vulnerable
to para-sats (for obvious reasons). The authors’ classification technology of
star/object/noise is only understandable in the context of para-sats. The
paper’s implied assumption of LEO object ephemeris determination using
star tackers is just wrong.

In the context of the paper, the RSO detection is based on star field images that would be captured without specific intent to do so, hence the opportunistic approach to the research. The focus of this research is on detecting and tracking general debris. Additionally, some of the future work enabled by this research is to study how thousands of star trackers, with potentially overlapping observations, can aid our space domain awareness.

If the authors re-write the paper dropping the wrong physics for the correct physics, it would make a very interesting contribution to sensor technology.

I'm not clear about the "wrong" and "right" physics classification made here. 

The authors also have to scrubb their references: for example reference 5
is https://amostech.com/TechnicalPapers/2020/Poster/Badura.pdf and reference 6 is https://orbit.dtu.dk/en/publications/space-debris-detection-andtracking-using-star-trackers.

These citations are relevant to the research presented, showing additional work in similar domain.

Reviewer 2 Report

In this paper the authors presented a star field image processing framework, RSOnet, using images from star trackers. Their innovation lies in using images from star tracker to identify celestial bodies, RSO, and noise for space domain awareness.

As to the technology framework, there are questions:

  1. A CNN is employed for object detection. But there are so many object detection methods, why the CNN is selected? Maybe some comparison should be done (with methods like SVM, random forest, deep CNN model and the transformer) to explain the advantages of the CNN model.
  2. the CNN model in Figure 3 is not so clear. the number of filters, the size of each filter should be listed. In my mind, for no useful feature can be extracted by a random convolution, the number of filters, the size of each filter are vital to their experiments. The accuracy of their CNN is no so high (in Table 1, 92%, 83% and 88% respectively), perhaps it can be heightened through precisely adjusting these parameters.
  3. the data used in experiments is not described in detail also. Maybe they can list the size, channels of the image, size of the whole dataset, the training and test dataset and so on.
  4. this paper has no enough comparison with other researches.
  5. some details:

1) In Fig. 3, why are these three convolutions getting smaller and smaller? In general, we often use an activation after a convolution, why there are 3 activation function listed together?

2) In Fig. 4, what is the meaning colors on the lines? Can they be labeled in the figure?

3) Page 6 section 4 line 6, “mean accuracy of 0.2 by epoch 30”. The value 0.2 should be 0.02?

4) The value in row Photon count (RMSE), column Synthetic of table 1 is “7 × 10--2”, is it a mistake?

Author Response

I have addressed all points brought forward by the reviewer. Additionally, I'd like to thank them for the questions and concerns, as they really helped me improve the explanation I provided for this work. The work presented in this paper is part of a much larger thesis, and it is difficult to explain one moving part amongst a system. Future work on this includes a multi-constellation analysis simulator looking at 1000s of star trackers for space domain awareness. Future work also includes hardware payload design and integration of the mentioned RSOnet architecture to optimize for detection and tracking in-orbit. Thank you!

In response to the reviewer's comments:

  1. A CNN is employed for object detection. But there are so many object detection methods, why the CNN is selected? Maybe some comparison should be done (with methods like SVM, random forest, deep CNN model and the transformer) to explain the advantages of the CNN model.
    I have now included a thorough comparison of other works in this field at the ends of section 2.1 and 2.2.
  2. the CNN model in Figure 3 is not so clear. the number of filters, the size of each filter should be listed. In my mind, for no useful feature can be extracted by a random convolution, the number of filters, the size of each filter are vital to their experiments. The accuracy of their CNN is no so high (in Table 1, 92%, 83% and 88% respectively), perhaps it can be heightened through precisely adjusting these parameters.
    I have updated the CNN architecture diagram Fig 3. I have also added more explanation to how it was designed, trained and validated in section 3.1 page 4.
  3. the data used in experiments is not described in detail also. Maybe they can list the size, channels of the image, size of the whole dataset, the training and test dataset and so on.
    I have added more explanation in section 4 page 7 regarding the training process, data used, amount of data used.
  4. this paper has no enough comparison with other researches.
    I have addressed some of this in the section 2.1 and 2.2, with further discussion in section 4.
  5. some details:

1) In Fig. 3, why are these three convolutions getting smaller and smaller? In general, we often use an activation after a convolution, why there are 3 activation function listed together?
The updated Fig 3 and accompanying text addresses some of this.

2) In Fig. 4, what is the meaning colors on the lines? Can they be labeled in the figure?
I have included a legend in Fig 2.

3) Page 6 section 4 line 6, “mean accuracy of 0.2 by epoch 30”. The value 0.2 should be 0.02?
Corrected. Many thanks.

4) The value in row Photon count (RMSE), column Synthetic of table 1 is “7 × 10--2”, is it a mistake?
Corrected. Many thanks.

Round 2

Reviewer 1 Report

The manuscript claims that star trackers can track low Earth orbit objects using multiple trackers. This is not possible mathematically. Ephemeris data can only be obtained by active interrogation of the object using either radio communications or radar/laser reflections. The readers of this manuscript would be deliberately misled into believing that this technology could be used for recovering previously seen orbiting objects or discovering new orbiting objects. The authors fail to see that ephemeris data is the ONLY data relevant for situational awareness. The reviewer stated that para-sats (close parallel orbiting objects) could be discovered using star trackers, but the authors never followed up on this remark. The ephemeris methodology in the paper is not supported by physics.

Reviewer 2 Report

As to question 4  in last review "this paper has no enough comparison with other researches".
I means the authors could compare their research with others. For instance, in Tab. 2 they can list tracking and classification performance of others for comparison.

Author Response

Please see attached PDF for changes made.

In Sections 2.2 and 3.1, additional comments and references have been added for comparison.

In section 4, pages 7-8, a more thorough comparison and discussion has been added to a newly added reference.

In section 4, pages 8-9, Table 2 has been upgraded to describe results of the algorithm discussed in further detail. Comparison to another new reference has been added in the discussion section. The comparison describes why the comparison is made, how they are different and what the results of this paper detail.

Thank you very much for your comments, they have helped the authors better address the shortcomings of the explanation provided in the paper.

Round 3

Reviewer 1 Report

Third referee report is included. If the authors do the two star tracker cooperative case mathematically, the paper will be accepted. This is the only criteria for acceptance.

Author Response

It is certainly not possible for a single star tracker to determine ephemeris of a low Earth orbit object, and no such claims have been made. The authors have exemplified a case study of a multi-observer scenario where two cameras are used, as explained by the reviewer, in a cooperative study. The case study and corresponding analysis is presented in the final section of the manuscript.

Please see the changes referenced to in the submission.